# Developing Pre-Service Teachers' Professionalism by Sharing and Receiving Experiences in the *Kampus Mengajar* Program

Suyatno Suyatno [1,*], Wantini Wantini [2], Dholina Inang Pambudi [3], Muqowim Muqowim [4], Agus Tinus [5] and Lilis Patimah [6]

1  Department of Education Management, Universitas Ahmad Dahlan, Yogyakarta 55166, Indonesia
2  Department of Islamic Education, Universitas Ahmad Dahlan, Yogyakarta 55166, Indonesia
3  Department of Primary Teacher Education, Universitas Ahmad Dahlan, Yogyakarta 55166, Indonesia
4  Department of Islamic Education, UIN Sunan Kalijaga Yogyakarta, Yogyakarta 55281, Indonesia
5  Department of Pedagogy, Universitas Muhammadiyah Malang, Malang 65145, Indonesia
6  Department of Islamic Education, Universitas Nahdlatul Ulama Surakarta, Surakarta 57141, Indonesia
*  Correspondence: suyatno@pgsd.uad.ac.id

**Abstract:** The low quality of teachers in Indonesia indicates that its current professional development programs are yet to reach an ideal form. Therefore, this research aims to determine how the *kampus mengajar* (teaching campus) program can be used to improve the professionalism of pre-service teachers to bridge the gap between theory and practice. Open interview questions were used to collect data from eleven pre-service teachers, two supervisors, and two principals involved in the *kampus mengajar* program with 3 months of professional experience placements in elementary schools spread across Indonesia. The data collected were analyzed using the thematic data analysis technique. The results showed that although the *kampus mengajar* program experiences various challenges, it can be used by pre-service teachers to enhance their professionalism by sharing and receiving experiences. The *kampus mengajar* program offers real and authentic experiences for pre-service teachers in guiding students, performing various administrative tasks, and learning in the workplace; thus, they have a better understanding of school life realities.

**Keywords:** pre-service teachers; professional experience; qualitative studies; sharing and receiving experience





## 1. Introduction

The low quality of in-service and pre-service teachers in Indonesia shows that its various professional development programs are yet to reach an ideal form. The World Bank reports that this profession was awarded low test scores on aspects of subject matter, including knowledge, pedagogic skills, and general intelligence [1]. Furthermore, motivation and dedication to teaching are low and associated with immature emotions and inadequate thinking [2]. Teachers in Indonesia are considered to have failed in carrying out their role in maintaining and improving students' learning abilities. As a result, student learning outcomes in Indonesia lag behind those of neighboring countries. According to the Ministry of Education's statistics, this inability is due to the teachers' relatively low level of academic qualifications [3]. In Indonesia, various professional development programs and other forms of training have been carried out, although these have been considered ineffective in improving teacher professionalism [4].

One of the main problems related to teacher professionalism is the gap between instilling theoretical and practical knowledge in the classroom [5,6]. There is a disassociation between program components organized by universities and the actual needs of elementary schools [7]. The gap between theoretical and practical knowledge also occurs because pre-service teachers are shaped by their personal experiences and opinions about this profession. According to Kertesz and Downing [8], this tends to cause dissonance

between universities and elementary schools. The lack of reciprocity results in a discord between pre-service teachers' preparation and the requirements [9]. Meanwhile, Darling-Hammond [10] highlights the disparity between the programs organized by campuses regarding field experiences. Although these are incorporated into the curriculum, the time spent by both pre-service teachers and their supervisors is often not properly planned. In addition, they are often left to work independently without guidance or supervision from the university [10].

Empirical evidence shows that field experience is an alternative as regards producing professionals [11–13]. It is an important opportunity for pre-service teachers to improve their skills [14]. This provides the best opportunity to learn and gain personal teaching efficacy after graduation [15]. For pre-service teachers, centers were provided for professional learning in the workplace to understand the daily realities of school life. Professional experience creates opportunities to harmonize ideas and theories learned at university. Field experience is a meaningful and practical teaching skill that is considered a basic need for pre-service teachers [16]. Deed et al. [17] stated that professionalism requires balancing the time devoted to theoretical learning at the university and the workplace. It was further clarified that there needs to be a meaningful integration of the two experiences to enhance professional outcomes. Pre-service teachers are expected to develop a teaching philosophy involving the transfer of theoretical knowledge acquired at university into an authentic learning context under real conditions. Teachers gain comprehensive and insightful constructive feedback from more experienced tutors who act as mentors [7,18]. Therefore, pre-service teachers need more practical opportunities on their way to becoming professionals [18].

The *Kampus Mengajar* Program is a subsidiary of the *Merdeka Belajar-Kampus Merdeka (MBKM)* or Freedom to Learn-Independent Campus policy. The MBKM is one of the programs initiated by the Ministry of Education, Culture, Research, and Technology (Kemdikbudristek) to provide field experience for pre-service teachers. This policy offers some form of autonomy to these institutions and freedom for students to select the desired program. It provides them with a broader learning experience and space to deepen their knowledge for three semesters, with two being spent off campus (equivalent to 40 credits, with 1 credit being equivalent to learning activities carried out by students for 50 minutes plus structured independent assignments relevant to achievement indicators specified in the curriculum) and one being in other study programs (equivalent to 20 credits). This includes internships, practical work, teaching assistantships, entrepreneurship, student exchanges, and research [19]. Moreover, this program is expected to improve the graduates' soft and hard skills competence to ensure that they are prepared for the dynamic needs of the classroom and as future leaders of the nation who are better qualified, moral, and ethical. An excellent leader is someone with adequate skills and experience needed to solve various challenges in the future [20]. This policy grants students the freedom to think either individually or in groups, thereby producing better qualified, critical, creative, collaborative, and innovative graduates in the future. The MBKM is expected to trigger their involvement in learning [21]. Moreover, it aids college students in exploring their greatest potential, including teachers, to independently improve the quality of learning [22]. For pre-service teachers, this program is expected to reduce the gap between the theoretical aspect learned on campus and the real needs of the school. It is assumed that there is a missing link between the courses offered by these universities and the actual needs of these users; therefore, graduates of the Teacher Training and Education Faculty are considered to be incompetent. Based on the research background, the problem lies in how the *kampus mengajar* program aids pre-service teachers in developing their professionalism, and its potential to bridge the gap between theory and practice are interesting aspects to be studied.

## 2. Literature Review

### 2.1. The Meaning of Professionalism and Teacher Professional Development Policies in Indonesia

The concept of "professionalism" is used to represent individual competencies and expertise as well as the quality of the work discharged [23]. It is based on three factors: using a specific scientific building, rendering services to others in the community, and possessing a self-regulated code of ethics to maintain high morale, quality, and ethical standards [24–26]. Professionalism has been defined in different ways, from training to personnel development to more efficient and effective professional courses [27–29].

Leung [30] categorized it into two types, independent and managerial. The first level or definition of this term, which is also called transformative professionalism [31], refers to how teachers view their practices, knowledge, beliefs, and skills, as well as critical reflection on past teaching experiences as learners, their abilities, and future directives. The second definition is managerial professionalism, which refers to what teachers expect as determined by official authorities, such as the Ministry of Education. In other words, the first level of professional development is a bottom-up, personal, and self-initiative process, while the latter is top-down, institutional, and other people-oriented [32].

These opinions showed that professionalism contains aspects of a person's expertise, skills, and abilities in specific jobs. It refers to the commitment in improving professional skills and developing strategies for carrying out work [33]. In Indonesia, the profession-alism of teachers is explained in Law No. 14 of 2005 concerning teachers and lecturers, which defines it as a job that requires specific skills and a decent source of livelihood. According to the law, teachers must have four competencies for professional work, namely pedagogic, professional, social and personality. The teachers' competence is a combination of knowledge and skills to perform effective learning and keep up with the times [34].

Several preliminary studies have identified the teacher professional development principles as being effective and successful. According to Lessing and De Witt [35], the three aspects include workshops, personal values and programs, and teaching approaches. Teacher professional development programs also need to enable them work with other colleagues to create organizations that support learning. They need to be given the opportunity to become practitioners, share knowledge and commitment, work with community members to implement coherent curricula and supportive systems for the students, as well as collaborate with them in ways that advance their understanding and skills. Teachers' professional development aims to improve their knowledge and skills through orientation, training, and support that enhance the quality of the teaching and learning process. It also focuses on core competencies such as improving their abilities, understanding the students, managing teaching skills and practice, knowledge of other disciplines, and appreciating their profession [2,35]. Preliminary research stated that professional development activities positively impact the teachers' beliefs and practices as well as students' performances in learning and educational reform in general [2]. These programs include relevant activities such as improved qualifications, updating tutors' knowledge and understanding of the subjects taught, practicing teaching students from different backgrounds, developing practical competencies and skills, learning new teaching methodologies, adopting learning innovations and technologies, improving ethical professionalism, as well as acquiring knowledge and skills to anticipate societal changes.

In Indonesia, professional development programs are carried out for both in-service and pre-service teachers. Several of them have been implemented for in-service teachers, such as PLPG (Professional Education and Training for Teachers), PPG (Teacher Certification Program), and SM-3T (Education by Bachelors at Frontier, Outermost, and Underdeveloped areas) in remote areas. Furthermore, they need to participate in various independent programs to develop their professionalism, such as subject–teacher consultations (MGMP). The Education Personnel Education Institution (LPTK) is the main initiative for pre-service English teachers in Indonesia, including those employed in state and private universities whose main role is to provide education and pedagogical training for those interested in teaching at both junior and senior high schools [36]. However, through professional teacher

education, the government is trying to improve the academic system by increasing their qualifications and skills.

Some research stated that the impact of teacher certification only improves their standard of living. Abbas [37] stated that the teacher certification program slightly contributed to improving the quality of national education. Teachers in Indonesia did not significantly improve their quality and performed poorly even after completing the certification program and receiving a salary increase. Helping them update their knowledge and skills in dealing with certain changes and managing human resources is needed. Additionally, professional development aids in achieving better performance in the aforementioned matter [2]. The essence of professional development is centered on teacher learning and transforming their knowledge into practice for the benefit of the students [38]. Walter and Briggs [39] reported that it is effective due to (a) the inclusion of concrete and classroom-based skills from outside the school, (b) teacher involvement in the selected fields to develop and carry out certain activities, (c) collaboration programs with peers, (d) providing opportunities for mentoring and coaching, (e) continuous and regular efforts, and (f) support for effective school leadership.

### 2.2. Professional Development for Pre-Service Teachers

One of the effective programs provides field experience for pre-service teachers. This helps individuals become tutors by learning the need for teaching with the support of a good mentor [40]. University-based pre-service teacher education is in a state of transition from a training model that emphasizes skill acquisition and competency mastery to a practice-based type centered on participation, involvement, and reflection [40]. This transition places importance on professional experience for pre-service teachers through practical field tasks in schools. The success of this transition requires collaboration between pre-service teachers' education providers in tertiary universities and schools [41].

Professional experience is relevant, and pre-service teachers typically describe it as the most important aspect of their program [42], partly because they value the opportunity to be mentored by experienced tutors during their teaching practice [43]. There is little doubt that effective mentoring is essential for the practical development of pre-service teachers in the workplace [44].

### 2.3. Kampus Mengajar Program in Developing Pre-Service Teacher Professionalism

The *kampus mengajar* program as a subsidiary of the Merdeka Belajar-Kampus Merdeka (MBKM) or Freedom to Learn-Independent Campus policy is one of the breakthroughs of the Ministry of Education, Culture, Research, and Technology (Kemdikbudristek) in terms of providing field experience for pre-service teachers. This autonomous academic institution and its complicated bureaucracy allow students to freely select the desired program. Furthermore, through this program, lecturers are given the freedom to develop their creativity and innovation in experience-based teaching, transform the curriculum according to regulations, and promote students to master various fields of knowledge useful for entering the world of work, without being limited by rigid and narrow administrative regulations. It is expected to improve the graduates' competencies, both soft and hard skills, to be relevant to the changing needs of the students and to produce future better qualified and morally ethical leaders [20]. This policy allows students to freely engage in critical thinking, thereby becoming creative, collaborative, and innovative. The *kampus mengajar program* is expected to increase students' involvement in learning [21,45]. In accordance with this program, college students explore their greatest potentials, including teachers, to innovatively and independently improve the learning quality [22]. For pre-service teachers, it is expected to reduce the gap between the theories learnt on campus and the real needs of the field. Moreover, it is assumed that there is a missing link between the courses taught at these universities and the users' needs, thereby producing incompetent graduates of the Teacher Training and Education Faculty.

According to the main pocket book of *kampus mengajar* guidelines [46], its flow is divided into three, namely the pre-assignment, assignment, and final assignments. The pre-assignment flow, which comprises debriefing and coordination, are activities carried out by students before partner schools. Partner schools are those designated by the Ministry of Education and Culture to assign participants to teach campus programs. Meanwhile, the assignment flow is divided into activities at the beginning, during, and at the end of the assignment. The beginning includes orientation, adaptation, observation, preparation, and consultation. At the orientation stage, students and the school introduce themselves, the group, and the person in charge. They also listen to the explanation from the school regarding its vision, mission, academic culture, school environment, as well as problems and challenges faced by the school. At the adaptation stage, students need to show an adaptive personality to be accepted by the school by being friendly, putting up feelings of appreciation, giving a sincere smile, paying attention to appearance and being willing to open themselves to learn and teach others. At the observation stage, students, with assistance from the school, conduct a series of direct observation activities by identifying the school's environment, administration, organization, and the learning process. The preparation stage is associated with the design of plans during *kampus mengajar* activities based on the results of observations. Finally, the consultation stage enables students to ask for approval of the design plan activity with accompanying teachers and field supervisors.

The assignment stage is divided into two major activities, namely teaching and non-teaching. Teaching activities carried out by students include the following steps: (a) Identify teaching materials according to students' abilities and the school's needs. (b) Design the teaching plan that will be implemented. (c) Discuss the plans made with colleagues and the school. (d) Reflect on learning activities conducted with colleagues and the school. (e) Make daily and weekly reports by filling out a daily logbook in the MBKM (Independent Learning-Independent Campus) application. Non-teaching activities are carried out by students in the following steps: (a) Identify personal abilities and school needs. (b) Design the non-teaching activities that need to be implemented. (c) Discuss the plans that have been made with colleagues and the school. (d) Reflect on non-teaching activities that have been carried out with colleagues and the school. (e) Make daily and weekly reports by filling out a logbook in the MBKM application. The non-teaching activities that can be carried out by students include: (a) Help the school to carry out administration of basic education-related data. (b) Carry out extracurricular coaching activities. (c) Implement guidance and counselling services with teachers. (d) Implement services for children with special needs. (e) Assist the teachers in carrying out administrative tasks [46]. The third stage enables the participants to carry out five activities: (a) Complete a self-assessment. (b) Request a peer assessment. (c) Complete a peer assessment. (d) Confirm filling in the assessment results carried out by the supervising teacher. (e) Compile the final activity report and upload it to the MBKM system according to the schedule [46].

*2.4. Weaknesses in the Literature and the Novelty of This Research*

Some research has been carried out on the *kampus mengajar* program. Based on the acquired results, there are at least three trends related to this matter, such as response to policies, the organizers' readiness, as well as its impact on students' skills. The research on the first trend was carried out by Wahyuni and Anshori [47], who examined the implementation of the *merdeka belajar* policy at Medan State University. It was reported that the students are aware of the importance of learning discourse on independent campuses, although some do not agree with the program. This research stated that low student literacy and lack of stakeholders' involvement were due to low student literacy. Students also believed that this program made it more challenging to graduate as planned. Other studies stated that most universities, especially those in remote areas, have not been in a hurry to adopt the program, and the unique situation of each presents certain obstacles during its implementation, such as students who lack understanding of the policy [48].

The second trend was carried out by Yusuf [49], who analyzed the relationship between the *kampus mengajar* program and university stakeholders' readiness. By adopting quantitative methods, the readiness of lecturers, students, and government support led to a positive impact on the enacted policy. Anwar [50] described the implementation of the *kampus mengajar* program at Muhammadiyah Elementary School 1 Padas. It was reported that in 2021 that teaching activities consist of face-to-face and online learning, including home visits. In addition, technological adaptation assists teachers in adopting teaching media and materials in accordance with the curriculum. Moreover, the administration carried out by the students was to aid in the preparation of learning tools as administrative completeness. The implementation of the *kampus mengajar* program, had a positive impact during the COVID-19 pandemic. Preliminary research stated that the *kampus mengajar* program triggers the students' readiness because it focuses on active learning, concept mapping, and value clarification based on information and communication technology [51]. This educational policy transformation is also in line with the academic theory and practice during the transition, stabilization, and growth potential periods [52,53]. Meanwhile, research on the third trend was carried out by Widiyono et al. [45], and it was reported that the implementation of the *Kampus Mengajar* program has a positive impact on students, such as triggering their interest in learning, as well as integrated literacy and numeracy skills. They also enjoy certain benefits, including being able to provide direct teaching experiences in order to develop interpersonal and leadership skills. Yohana et al. [54] examined the factors that influence the entrepreneurship program in the Kampus Merdeka policy by using five universities as the research objects. It was reported that the campus policy factors, apprenticeship programs, and exploration of local potentials positively affect competency development, implementation, and entrepreneurial learning.

A review of the previous studies shows that there is little or no interest in analyzing the professional experiences gained through the *kampus mengajar* program. This is because it had only been running for one year. Therefore, this research has the potential to develop an alternative model to develop the professionalism of pre-service teachers. Based on the theoretical framework, this research aims to explore: (1) what experiences are obtained by pre-service teachers during the *kampus mengajar* program to improve their professional experience, and (2) how the program can be utilized.

## 3. Materials and Methods

### 3.1. Research Type

The qualitative method was used to obtain detailed information about the complex phenomena of campus teaching programs and identify themes and theoretical structures that describe the process [55]. This method allows for a better understanding of human behavior, with a focus on naturally occurring phenomena and their complexity. It also allows for individuals to have an in-depth understanding of these phenomena [56]. Descriptive data in the form of email conversations and open-ended questionnaire interviews with participants were used to conduct this research. These data were then analyzed using one of the qualitative data analysis techniques, namely thematic analysis. Informed consent was given by the participants who were reminded that: (1) their involvement was voluntary, (2) all data collected are de-identified when used to draft publications, and (3) they are not required to answer unwanted questions. Moreover, all their names were written using the code P1–P15 to ensure the confidentiality of their identities.

### 3.2. Participants

The research participants were purposively selected from students who are members of the WA group, a campus teaching program at a university in Yogyakarta. From a total of 55 students, 13 were voluntary participants, while 2 eventually withdrew. In the second stage, the 11 students were placed in five different schools with field supervisors. In the third step, of the five field supervisors and five school principals where the students were deployed, only two each stated their voluntary willingness to become participants.

The rest were not interested due to tight schedules, which could prevent them from granting interviews according to a predetermined deadline. Smith et al. (2009) stated that a small sample size is acceptable because qualitative is an ideographic approach related to understanding certain phenomena. They were determined by the sampling techniques and fulfilled the following criteria: (1) involved in the *kampus mengajar* program, (2) were science students (pre-service teachers), (3) placement in elementary schools, (4) volunteered to participate in the research until it was completed. Overall, the participants consisted of 13 women and 2 men. The participants from the pre-service teachers, field supervisors and school principals are in categories of P1–P11, P12–13, and P14–15.

### 3.3. Research Procedure

Before data collection, a letter of permission was requested from the university's vice-chancellor, where the students participated in the *kampus mengajar* program. Data collection was conducted in two stages. The first stage was carried out through email conversations, a technique rarely used to generate rich data when carried out appropriately in qualitative research [57–59]. The first stage was conducted by sending a list of questions through email to the participants, and they were given a maximum of two weeks to complete the forms. Based on the allocated time, 13 of them responded on time, while the remaining 2 responded in the next 2 days after receiving confirmation. The answers obtained through email were re-transcribed, and the participants' answers were read in their entirety to get an overall idea. In the second stage, in-depth answers were collected from participants through virtual live interviews using zoom meetings. A total of seven identified participants stated their ability to hold a second interview, consisting of five pre-service teachers and two field supervisors. Interviews with open-ended questions were carried out to obtain information from participants for proper conveyance of experiences without being limited by the perspective of previous findings. Open-ended answers to questions allow participants to create options for responding. Not only did this provide an opportunity for participants to share their experiences but also for us to gain more insight [60].

Interviews were carried out based on certain guidelines and performed once or twice with different participants. Each selected the time according to their desires, with the duration lasting for a minimum and maximum of 22 min and 56 min, respectively. Interestingly, each was recorded and transcribed verbatim afterward. Data validity was ensured by comparing the findings of the first, second, and third authors, as well as the reviews of the fourth and fifth. The three authors mentioned earlier analyzed the raw data by creating categories through themes and identifying patterns in a similar way.

### 3.4. Data Analysis Technique

The thematic analysis technique for developing themes in the form of patterns was used to analyze the data collected [61,62]. This process was conducted in two stages: the first opens the code of raw interview data [63] to identify instances when participants described the use of the *kampus mengajar* program to enhance teachers' professional experience; the second is the analytic coding stage, which is conducted by connecting and solving various similar codes to obtain conclusions from the first stage [63]. The data analysis process is manually organized, without the help of software. In developing the themes, the authors provided support by using descriptive, linguistic, and conceptual comments. According to Yin [64], field notes or memos are used to support all data analysis activities, which guides data reduction (extracting the essence), presentation (organizing meaning), and drawing conclusions (explaining findings) [65]. Four important themes were found through this process, namely (1) sharing experiences supported by four sub-themes, such as building motivational and varied learning opportunities, (2) receiving experiences supported by three sub-themes, (3) developing professional abilities supported by three sub-themes, and (4) challenging programs supported by three sub-themes.

## 4. Results

This research aims to explore the pre-service teachers' experiences gained from the *kampus mengajar* program to enhance their professionalism. Data analysis produces four themes, such as sharing and receiving experiences, professional development ability, and program challenges. The data analysis results are shown in Table 1.

**Table 1.** Data analysis results.

| Theme | Sub-Theme | Number of Participants |
|---|---|---|
| | Building motivational and various learning opportunities | 15 |
| Sharing experience | Helping school administration | 10 |
| | Helping teachers adapt to technology | 12 |
| | Developing school programs | 10 |
| | Being trusted | 12 |
| Receiving experience | Getting the opportunity | 11 |
| | Obtaining real experiences | 7 |
| Professional skills' development | Enhancing soft skills | 12 |
| | Trained skills | 10 |
| | Time management | 12 |
| Program challenges | Program socialization | 12 |
| | Coordination | 15 |

### 4.1. Sharing Experience

This theme implies that while participating in the *kampus mengajar* program, pre-service teachers had the opportunity to share their experiences, skills, and knowledge with the schools where they taught. It was supported by 100% of participants, and it is divided into several sub-themes, including successfully guiding the students, helping the school administration and teachers adapt to technology, and developing school programs.

#### 4.1.1. Building Motivational and Various Learning Opportunities

Pre-service teachers' experience in successfully building motivational and various learning opportunities is an interesting one, particularly during the *kampus mengajar* program. This is manifested in the form of fun learning in class, to motivate the students and ensure they do not get bored easily, as well as being accompanied by home visits. All participants (n = 15) reported this event during the data collection process, as shown in the following comments.

> "I taught citizenship education, namely how to practice Pancasila daily. They happily participated in learning". (p2, 10 to 13)

Similar comments were also conveyed by many other participants: "I have succeeded in assisting students through the home visit program" (P4, 3 to 5); "I developed the subject matter from thematic books to be more creative and ensure the students are not easily bored and absorb the lessons more quickly" (p2, 88 to 90). The pre-service teachers' abilities to accompany these learners are evidence of their successful experiences enhancing their professionalism.

> "The valuable experience I gained during the *kampus mengajar* program was being able to provide varied learning where previously students studied with the teacher only with the lecture method, but now I use the learning method by inviting students to play roles directly". (P2, 3–8)

P2 adopted role-play learning and it made the students happy because teachers at school rarely applied this method.

### 4.1.2. Helping the School Administration

The experience gained from helping with the school administration included setting exam questions, report cards, and supervising these activities. P5 stated that during the *kampus mengajar* program, they often engaged in helping school administration. "P5 personally executed this activity because the existing teachers were not used to the use of technology, such as laptops". Based on the fact that none of them were capable, P5 was forced to share the experiences gained and this led P5 to completely understand the duties. "P5 realized that the teacher's job was not only teaching". (P5)

Similarly, P1 reported, as follows:

> "My most valuable experience was when my colleagues and I helped the teacher council prepare for school exams, starting with typing the questions, making report cards, and participating in the supervisory activities". (P1, 3 to 6)

Other participants also shared their experiences:

> "Some also helped the administration by stamping books stored in the warehouse because the school does not yet have a library, and by assisting the teachers' needs". (p2, 17 to 20)

### 4.1.3. Helping Teachers Adapt to School Technology

During the online learning stage implemented due to the COVID-19 pandemic, technological adaptation was a major problem faced by virtually all teachers. The sudden demand caused the majority not to have enough time to improve their ability to use technological devices in learning, as stated by one of the participants below:

> "My friends and I held a workshop to introduce a "Canva" design application to facilitate teachers in creating learning media, certificates, banners for school activities, or concept maps. In addition to that, my friends and I also held a workshop on Google Workspace consisting of Google Classroom, Google Meetings, Google Forms, and Google Drive. This was based on the fact that initially, only a few of them were able to use Google Workspace to support distance learning". (P4, 16 to 24)

The story of P4 shows that the majority of the teachers are not yet proficient in using technology to assist with online learning. This condition provides an opportunity for pre-service teachers to share the strategies needed to access various platforms in online learning. "The program organized by P4 starts with adapting to zoom meetings, Google Meet, Microsoft Excel, and how to scan using a cellphone" (p1, 15 to 17). A similar story was also reported by P3, as follows:

> "I discovered that the teachers used manual (handwritten) report cards, which was quite different from the surrounding schools that had switched to the automatic type with the help of Microsoft Excel and similar applications or programs. This prompted me to design automatic report cards using Microsoft Excel to help the school adapt to technology. They received positive responses from the teachers, students, and parents". (P3, 7 to 12)

### 4.1.4. Developing School Programs

Pre-service teachers also shared certain experiences in developing school programs, such as making webinars, holding national day commemoration competitions, engaging in home visits, and forming study groups. One of the participants stated that they engaged in home visit programs because it was discovered that some students did not have the motivation to learn while studying from home.

> "Home visit programs are for students who have no motivation to learn. They are guided and assisted until they become active in learning". (P4, 9 to 12)

Additionally, the formation of study groups was also needed during the pandemic because it aids students who have learning difficulties. "I created study groups for lower-grade students and guide them in their reading" (P5, 18 to 20). Another participant explained as follows:

> "Another valuable experience was when my friends and I made a national webinar themed 'Improving Teacher Competence through Fun Learning and Dancing in the New Normal Era.'. My friends and I were very happy because there were many enthusiasts in the form of school teachers who wanted to take participate, and in a short time, it was completed without any obstacles". (P1, 7 to 11)

### 4.2. Receiving Experience

The theme receiving experience simply means that during the *kampus mengajar* program, the participants acquired knowledge and skills from the school, especially as regards learning directly from their teachers. Likewise, in sharing experience, 100% of participants also reported sharing experience while attending the *kampus mengajar* program. The sub-themes are being trusted, obtaining the appropriate opportunity, and gain real experience.

### 4.2.1. Being Trusted

Most of the participants (n = 12) enthusiastically shared how they were trusted by the school to carry out various activities, either in the form of hosting an event, contributing ideas for its development, or the provision of teaching materials. P1 was entrusted with hosting the graduation ceremony. "P1 felt more confident because she properly executed the task" (P1, 43 to 45). In line with P1's story, it has been stated that:

> "The classroom teacher trusted them to provide teaching materials in ICT learning and help assess the students during practice. This experience gave me a clearer picture of tomorrow when I become a teacher as regards developing learning materials". (P7, 24 to 29)

### 4.2.2. Obtaining the Opportunity

Obtaining the opportunity during a *kampus mengajar* program enhances pre-service teachers' professionalism. The participants shared many stories about accompanying the students during competitions, giving speeches in a series of events, especially in the Ramadhan month, aiding children with special needs, and contributing ideas for school development. P6 reported his experiences when he had the opportunity to assist students in various competitions and succeeded in leading them to win at the regency and provincial levels.

> "At the end of the *kampus mengajar* program, I was happy because the students I mentored won the competition. Meanwhile, at the sub-district level, those who participated in the dance and weaving competition each won 3rd place. They were also invited at the regency level, and alhamdulillah the student who took part in the weaving competition, won 1st position while those that participated in dancing won 2nd place. All thanks to the teachers who cooperated, the students are able to participate in the weaving competition. In addition, I am excited, touched, and proud". (P6, 7 to 13)

P5 reported that their trust in him when he attended various meetings to convey ideas made him feel valued.

> "During the meeting, I was also allowed to express my opinion regarding any experience during my time at the school. I feel appreciated by the teachers irrespective of whether I am still a student". (P5, 56 to 58)

Similarly, P7 reported that accompanying students with special needs was a valuable opportunity for him to understand their learning characteristics. "The opportunity was given to me by the class teacher to accompany them directly. Therefore, I was privileged to

understand how much students learn" (P7, 35 to 37). On another occasion, P6 also reported that "I was allowed to give a speech for 2 days at a short-term Islamic boarding school activity during the Ramadhan month. At first, I was embarrassed; however, thank God I was able to deliver it properly" (P6, 40 to 43).

### 4.2.3. Obtain Real Experiences

The *kampus mengajar* program has provided best practices for pre-service teachers, such as carrying out learning, increasing students' motivation, engaging in home visits, working selflessly, being a parent figure, teaching sincerely, giving rewards, not discriminating against students, and properly managing the school program. Interestingly, more than half of the participants recounted this experience (n = 9). P3 described how she obtained real experiences in managing character education and strengthening certain programs.

> "I discovered an interesting thing about religious character education. The school is good at implementing this subject, and the parents really appreciate this, which also amazes me. This religious character is applied based on ahlussunah wal jama'ah". (P3, 14 to 18)

In contrast to P3, who gained several experiences in managing a school program, P4 describes how she obtained real experiences from this learning activity.

> "I gained experience from the teachers in terms of executing the learning process. In addition, I am aware of how the teachers continuously motivate the students by properly delivering the learning materials and communicating with them". (P4, 43 to 47)

A similar experience was also shared by P4 and P7. P4 witnessed how the teachers visited the students' respective homes. "I came to understand how they are guided selflessly, regardless of their background." Furthermore, "I also understood how to act as a second parent figure (P4, 78 to 82). P7's story reinforces this subtheme: "I gained certain experiences from the classroom teacher about how to give attractive rewards and teach sincerely without discriminating" (P7 92 to 95).

### 4.3. Professional Skills Develop

The sharing and receiving experiences gained by pre-service teachers during the *kampus mengajar* program aided them to develop pedagogic, professional, personal, and social competencies needed to boost their professionalism. This theme is formed by two sub-themes, soft and honed skills, as follows.

### 4.3.1. Enhancing Soft Skills

Soft skills are one of the important abilities that need to be embraced by prospective teachers in terms of carrying out their duties professionally. These aid in establishing communication, maintaining good relationships with colleagues, students, and the surrounding community, and being inclusive and developing emotional maturity, tolerance, and social sensitivity. The *kampus mengajar* program has enabled the research participants to acquire all these attributes. An interesting story was told by P8:

> "I learned how to communicate with teachers from different religious backgrounds. In this school, the students were of two beliefs, Hinduism and Islam. Based on my observations, the majority of Islamic students often discriminated against those with different beliefs. I was forced to teach them how to tolerate one another. Therefore, I developed the habit of reading stories related to tolerance in the mornings". (P8, 46 to 55)

P2's participation in the *kampus mengajar* program improved "my communication, leadership and self-confidence skills" (P2, 69 to 73). Similarly, P10 stated that "my personality was developed, especially in the aspect of discipline, and ensuring there is harmony among my peers, mentors, and students" (P10, 40 to 43). For P1, this program triggered

some attributes. "My social sensitivity is getting higher because I often communicate with teachers, parents, and all the students" (P1, 53 to 56, P3, 59 to 60). Meanwhile, P8 stated that schools outside Java lead her to be concerned about the existence of inequality in this country.

> "Interacting with principals, teachers, and students made me realize how visible the gap is between these institutes and the elementary ones during my internship in Yogyakarta". (P8, 3 to 6)

### 4.3.2. Skills Are More Honed

One advantage of field experience is that pre-service teachers have the opportunity to put their knowledge into practice, although this is not necessarily acquired while studying on campus; therefore, their skills are honed, as stated by P9.

> "While following the *kampus mengajar* program, I felt happy to be able to teach elementary school. What my friends and I gained from it was not obtained on campus". (P9 4 to 11)

The opportunity to instill knowledge in the real-world context tends to ultimately hone the skills of pre-service teachers, as reported by P2.

> "The *kampus mengajar* program serves as a forum to practice my skills, gain experience, and turn it into an extraordinary lesson". (P2, 138 to 141)

### *4.4. Program Challenges*

Although the *kampus mengajar* program offers several benefits for pre-service teachers to enhance their professionalism, it is undeniable that it has numerous challenges that need to be overcome. All participants (n = 15) shared this experience. Additionally, this theme is formed by three sub-themes, such as time management, program socialization, and coordination between organizers.

### 4.4.1. Program Socialization

As a new initiative by the Ministry of Education and Culture (launched in early 2020), program socialization is a major problem in implementing *kampus mengajar*. Moreover, all participants complained about this issue (n = 15). The lack of socialization caused this program not to run optimally. P11 stated: "Hopefully, the *kampus mengajar* needs to be socialized because many students still do not know about this program" (P11, 88 to 90). The other participants had similar experiences, as recounted by P2.

> "Many students who participated in the *kampus mengajar* program are still not aware of the materials to be taught in elementary schools. Moreover, they do not know how to maximize their objectives". (P2, 160 to 163)

P5 stated a similar experience, as follows:

> "The socialization of *kampus mengajar* needs to be expanded, assuming it is possible, because many students are still not aware of this program". (P5, 76 to 80)

### 4.4.2. Coordination between Organizers

Lack of socialization has caused most of the parties involved in *kampus mengajar* not to have the same perception about this program. Therefore, there is a need for coordination between organizers. This challenge was also shared by all participants (n = 15). P11 recounts her experience, as follows:

> "*Kampus mengajar* should be socialized in schools that have been selected as partners to ensure there are no misunderstandings because many equate this program with internships and real work lectures (KKN). Meanwhile, whenever my friends and I, who are participants, do not come to class, as usual, one of the teachers says, "you need to teach?" even though this program focuses more on literacy and numeracy". (P11, 92 to 98)

Other participants also discussed the lack of coordination between the organizers, which resulted in the selection of schools that did not fulfill the criteria. Based on the interviews held with P7 and P12, the following was reported:

"The selected schools really need to be observed to ascertain whether these deserve to be improved or assisted through student creativity, aspirations or attention of the organizers in terms of realizing program". (P7, 114 to 117)

"The *kampus mengajar* program socialization needs to be re-expanded because individuals, such as the schools where I was assigned to, are not aware of its existence". (P12, 33–37)

P12's experience shows that the idea or initiative led by the Ministry of Education, Culture, Research, and Technology regarding the recognition of the *kampus mengajar* program and its division into course credits has not been fully understood by the host universities.

### 4.4.3. Time Management

Time management is also one of the themes that were complained about by many participants, including being out of sync with the learning schedule at school, or the university, which is considered less specific; therefore, its utilization becomes ineffective. P1 reported that "I felt that the allotted time was inappropriate because when the school lesson was over, I had just been sent to the field, and this was regretted by many teachers" (P1, 83 to 85). This issue also occurred because the schedule for the assignment and that of the campus also coincided. This is burdensome for the participants because the majority are undergraduates. "As a result, I missed a few courses on campus, and for college, I had to work hard because I have a huge responsibility" (P11, 83 to 85). The division of the *kampus mengajar* program schedule was also an obstacle for some participants, as explained by P14.

"There is a need for more detailed scheduling of the objectives to be achieved during the assignment, such as helping out with the teaching process, including administration and technological adaptation. Therefore, the intention of the Ministry of Education and Culture as the goal of the *kampus mengajar* program is carried out properly". (P14, 83 to 86)

## 5. Discussion

The main findings of this research show that pre-service teachers benefit from the *kampus mengajar* program by sharing and receiving experiences to enhance their professionalism. They share by successfully building motivational and varied learning opportunities, helping school administrators and colleagues to adapt to technology and through other activities. Meanwhile, receiving experience was obtained because pre-service teachers are trusted by the school and gained real experience on various educational programs that have been theoretically studied on campus. In general, this research showed that the placement of pre-service teachers in schools has authentic and real experiences, with mutual benefits between schools and universities.

The experience gained by pre-service teachers during their 3-month assignment is an important opportunity to improve the required skills. This centers on professional learning at the workplace, where they understand the daily realities of school life. Professional experience provides opportunities for pre-service teachers to harmonize ideas and theories that were learned at the university [16]. They are expected to develop a teaching philosophy by transferring the acquired theoretical knowledge into an authentic learning context under real conditions. This process gained comprehensive and insightful constructive feedback from more experienced teachers perceived as mentors [7,18]. Therefore, pre-service teachers need more practical opportunities to boost their professionalism [18].

However, becoming a professional teacher requires balancing the time devoted to studying various theories learned at the university and learning in real-world contexts. There is a need for the meaningful integration of these two experiences to improve prospec-

tive teachers' learning and professional outcomes [17]. Interestingly, this sharing and receiving of experiences is also described by Brante [66] regarding the need for a link between scientific theory and professional practice. It was referred to as a "dialectic between know-why and know-how, based on a shared platform of science and profession" [66]. According to Brante, the meeting between practitioners and scientists is relevant for both parties because the model quality depends on being "developed, modified, and occasionally rejected by input from both parties, namely from a scientific or theoretical and professional or applied aspects." In this research, practitioners mentor these teachers when they are given assignments. The *kampus mengajar* program offered real and authentic experiences in guiding students, performing administrative tasks, helping their colleagues adapt to technology, and developing various academic tasks. Furthermore, they also gain trust, opportunities, and best practices from their mentors in studying the professional world of learning. Scientists are played by field supervisors as well as the theories learnt on campus. All these processes are packaged in sharing and receiving experiences. It is assumed that the "picture of the subject matter is perceived as a shared perspective of basic causal mechanisms" [66].

The sharing and receiving of experiences led to the development of professional competencies. In this research, it was reported that by participating in the *kampus mengajar* program, these teachers developed soft and honed skills components. These results reinforce previous findings [8,67,68], which stated that collaboration and partnerships between schools and universities tend to support pre-service teachers in terms of improving their professionalism. During the placement process, their mentors play a vital role in guiding and boosting their growth [69]. In order for this process to be effective, these mentors are expected to possess good communication skills and clearly articulate each party's roles. Therefore, the selection process was based on expertise and not just as a matter of seniority [12]. In the campus teaching program, tutors are teachers appointed by the principal to assist participants during assignments at school. The tutors' selection is based on their competence in terms of pedagogic, professional, social, personality, use of information technology in learning, and school management skills.

Although the *kampus mengajar* program aids participants in developing their professional experience, it was observed that it still left some managerial challenges, such as time management, program socialization, and coordination between organizers. Hasty time management causes pre-service teachers to find it difficult to reconcile their various activities at the school and on campus because these coincide. This is exacerbated by the socialization of the program, which is still not optimal. Pre-service teachers, mentors, principals, supervisors, and universities have different perceptions. Therefore, the coordination between the Ministry of Education, Culture, Research, and Technology (Kemendikbud-ristek) as the program's host, universities' supervisors and schools needs to be improved. This condition also aligns with the various challenges encountered in previous field experiences. Hoffman et al. [41] stated that the lack of coordination between the mentors at schools and supervisors at universities contributed to the substantive support of pre-service teachers.

Generally, these findings indicate the urgency of a stronger theoretical framework on the linkage between the Ministry of Education, universities, and schools as a whole to be more proactive and responsible in producing prospective professional teachers. This is also in line with the post-practical method paradigm where teachers are no longer considered as consumers of theory [70]; rather, they are perceived as constructors. The pattern is also relevant to recent studies that propose that professional development is a bottom-up process [32]. This implies that appropriate professional learning is realized through various "differentiated, and contextualized stages, related to practical, curious, collegial, and collaborative problems." Furthermore, it represents an active process that shapes and promotes the teachers' learning skills [71]. Dewey's experiential theory creates meaningful experiences while engaging in the teaching profession [72]. This allows pre-service teachers to translate the basics of theoretical courses into practical learning activities in the classroom [73]. Theories learned in universities from reading and analyzing texts,

lectures, tutorials, and discussions are also encountered through teaching practice in authentic settings to minimize the gap between this hypothesis and practice [74]. Contextual involvement of pre-service teachers is important. Burns [75] stated that "teacher learning is not perceived as translating knowledge and theory into practice, rather as an effort to embrace new ones by participating in certain social contexts, activities, and processes. This is sometimes called "practitioner knowledge," the primary source of practice and understanding for teachers.

In Indonesia, the time given for the field introduction program of previously existing pre-service teachers in internships/practice field is only 1 month. This program is considered unable to provide authentic experiences to students due to the short assignment time, monotonous task dimensions, and inadequate supervision from university supervisors at the school. Meanwhile, programs to increase teachers professionalism in the form of Professional Education and Training through certification programs are also considered ineffective [76]. One of the alternatives to introducing pre-service teachers to a more challenging real-life school, diverse peer collaboration experience, varied programs, and longer school assignments is by using the *kampus mengajar* technique.

This research successfully revealed how the *kampus mengajar* program provides real experiences to pre-service teachers by sharing their experiences to enhance their professionalism. The experience gained by teachers is used to guide students, perform school administrative tasks, assist colleagues in schools in adapting to technology, develop various academic tasks, and provide a workplace conducive to learning. However, this research was unable to reveal how the *kampus mengajar* program can improve the hard core of pre-service teachers' teaching quality because the focus was on the "assignment" stage [46]. The last stage determined the quality through four important activities, namely completing the self-assessment, requesting and completing peer assessment, and confirming the completion process [46].

## 6. Conclusions

This research discovered that the *kampus mengajar* program is utilized by pre-service teachers to share and receive experiences. With this reciprocal process, all parties benefit from the process. Mentors and school teachers tend to upgrade their skills relating to managerial practice and classroom learning by sharing insights and theories. Conversely, pre-service teachers improve themselves because they have the opportunity to experience real school life and acquire real experience, including learning from their mentors. This research has several limitations: First, only pre-service teachers, principals, and supervisors involved in the *kampus mengajar* program were interviewed. Second, the characteristics of the curriculum implemented in the elementary schools are different from those implemented in the next level, which led to the conclusion that this research tends not to be generalized. Furthermore, due to technical limitations, data collection was realized using a single technique: interviews. Third, this research did not compare the effectiveness of the campus teaching program with that accepted by pre-service teachers in enhancing their professional experience. This created an opportunity for future studies to explore this problem from various perspectives. Subsequently, there is a need to ascertain how this *kampus mengajar* program is utilized by pre-service teachers to develop their professionalism by involving various participants and more diverse data collection techniques. Further research capable of comparing the effectiveness of campus teaching programs with other pre-existing ones is needed for the continuous evaluation of various programs from the Ministry of Education. It is also important to determine how the *kampus mengajar* alumni adapt to the demands of the real working-class world, such as the significant differences in the readiness of alumni and non-alumni teachers.

**Author Contributions:** Conceptualization, S.S., W.W. and D.I.P.; methodology, S.S., W.W. and D.I.P., validation, S.S., W.W. and D.I.P.; formal analysis, S.S., W.W. and D.I.P.; investigation, S.S., W.W., D.I.P. and L.P; resources, S.S., W.W., D.I.P., M.M. and A.T.; data curation, S.S. and D.I.P.; writing—original draft preparation, S.S. and W.W.; writing—review and editing, S.S., M.M. and A.T.; visualization, S.S.;

supervision, M.M. and A.T.; project administration, S.S. and D.I.P.; funding acquisition, S.S., L.P. and W.W. All authors have read and agreed to the published version of the manuscript.

**Funding:** This research was funded by Universitas Ahmad Dahlan, grant number PD-059/SP3/LPPM-UAD/VII/2022.

**Institutional Review Board Statement:** All subjects gave their informed consent for inclusion before they participated in the study. The study was conducted in accordance with the Ethics Committee of Universitas Ahmad Dahlan Indonesia.

**Informed Consent Statement:** Informed consent was obtained from all subjects involved in the study.

**Data Availability Statement:** Not applicable.

**Acknowledgments:** The authors are grateful to the Universitas Ahmad Dahlan, UIN Sunan Kalijaga Yogyakarta, Universitas Muhammadiyah Malang, and Universitas Nahdlatul Ulama Surakarta for supporting this research.

**Conflicts of Interest:** The authors declare no conflict of interest.

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
