# Peer review of "Developing Pre-Service Teachers’ Professionalism by Sharing and Receiving Experiences in the Kampus Mengajar Program"

_education, doi:10.3390/educsci13020143_

Round 1
Reviewer 1 Report
Thank you for the opportunity to review the manuscript ”Developing Pre-Service Teachers' Professionalism Through Sharing and Receiving Experiences in The Kampus Mengajar Program”. The goal to improve teaching quality in Indonesia by developing preservice teachers’ field experience is very important and these efforts deserve to be investigated. I am suggesting a major revision, however, before the manuscript can be accepted for publication.
Theoretical background
The theoretical part describes the concept of teacher professionalism (i.e., three essences, independent/managerial), but never define explicitly how it is understood in this study. I suggest further elaborating the concept of teacher professionalism, the main concept of the study.
Investigating the utility of the Kampus Mengajar Program in developing preservice teachers’ professionalism appears to be the main goal of the study. The specific research questions are lacking, however, and I am asking the authors to provide them at the end of the theoretical part.
Given this goal, it was surprising that a strong summary of the specific activities, guidelines, or contents of this program was lacking from the introduction. The authors do describe the goals of the program (e.g., to improve the graduates soft and hard skills competences) but do not provide details about this specific programme. How structured was it for the preservice teachers? Was the method of implementation freely up to students and schools/supervisors/principals to decide? What concrete actions it required from the students and what from the supervisors or the school? How were they monitored? What specific modules or activities support the links between theory and practice? Was the improvement in preservice teachers’ literacy and numeracy skills a concrete goal of the program, what activities helped to achieve this goal, and how it was evaluated?
Method
The authors claim to use phenomenological approach, which I find problematic, however. I am wondering is this phenomenological analysis or basic thematic analysis?
First, it is somewhat unclear what is the phenomenon of interest. Is it preservice teacher’s field experience and what it means to him or her? Yet, the data also includes data collected from supervisors and principals. How would this data help us understand the very personal lived experience of a preservice teacher?
Second, an important part of data appears to be written replies to a list of questions sent via email to the participants. This information was obtained from all 15 participants, whereas only 7 participants participated in interview via virtual meeting. It is unclear who were the participants in the interviews, presumably a combination of preservice teachers, supervisors and principals? Please provide this information.
It seems unlikely to me that written replies to an email would produce data revealing the in-depth knowledge of the phenomenon (presumably the lived experience as a preservice teacher in a school) and what it means to a person. In my experience, written questions often are relatively short and superficial reports. At the very least, researchers should provide information on what the participants were asked and how long the answers were in written form.
In sum, I am asking the authors to describe in more detail why they think this is a phenomenological study, what is the phenomenon they aim to study, and how different parts of the data contribute to the analysis and findings. Alternatively the authors could consider using, for instance, a traditional thematic analysis to identify the essential topics or themes forming the data. It seems to me that the findings have been reported following the style that is typical to thematic analyses rather than to a phenomenologial study.
Results
I fear that giving the names of the schools where preservice teachers taught risks the anonymity of the study participants. Protecting the identity of the participant is an important data protection principle, and therefore I would definitely leave out the names of the schools, as they do not affect the interpretation of the results in any way.
I am also wondering, do the authors also teach the participants, as this might influence how the participating preservice teachers evaluate the program. Please specify your relationship with the preservice teachers.
One subtheme ”successfully guiding students” could perhaps be ”building motivating and varied learning opportunities”, because it was mostly about planning teaching that was fun and motivating, and included new teaching methods.
One subtheme ”best practices” raises the question of whether the quality of practices in the school were evaluated before allowing the students enter the field experience. How sure can one be that the school was using the best practices in the field?
One subtheme ”soft skills develop” describes that the program has enabled the research participants to acquire many favourable attributes. I would recommend that the expression be softened, because it was the participants' own assessments of the development of their skills, not objectively verified increase in skills.
Overall, I feel that the Results section does not meet the requirements of a phenomenological analysis as it fails to provide a in-depth knowledge of particicants’ experiences in the field and what is actually means for them.
Discussion
The discussion focuses on considering the program's contribution to the connections between theory and practice. The authors argue that ”this research explicitly reduces the gap between professional development’s theoretical and practical aspects” and ”professional experience provides opportunities for preservice teachers to harmonize ideas and theories that were learned at the university”. I fear that these are overstatements of the findings. None of the themes in Results section described how the educational concepts learned at the campuses turned into concrete actions or teaching in the field experiment or how the preservice teachers were reflecting the links between theory and practice. Instead of assuming, a stronger empirical evidence is needed to show that the gap between theory and practice was reduced.
Instead, in my view, the findings are in line with the statements that ”program offered real and authentic experiences in guiding students, performing administrative tasks, helping their colleagues adapt to technology, and developing various academic tasks” and included ”learning at the workplace, where they understand the daily realities of school life.”
Judging on the basis of the findings, it seems that the program gave participants much needed experience of school life and promoted self-assessed soft skills but did not succeed well in fostering the ”hard core of teaching quality”, that is, the quality of classroom interaction. Alternatively, it is possible that the interview questions and resulting data did not require the participants to reflect these latter skills that are crucial for effective learning. I would like to see discussion about how the authors would improve and develop the program further.
Minor:
- Please specify what PLPG, PPG, and SMT (line 125) mean.
- The manuscript states that all names were written using the codes R1-R15. Codes included P rather than R?
- It would be helpful to be able to identify whether the participant providing a citation is a preservice teacher, supervisor or principal.
- Language check is needed.
- Supplementary Materials were not available at www.mdpi.com/xxx/s1.
Author Response
Dear reviewer
Thank you for your suggestions. We are very happy to receive feedback from reviewer described in our article. We have work hard to respond and revise our article based on suggestions from reviewer. Here, we attached the revised version by article track changed.
Kind regards
Reviewer 2 Report
The article deals with a very important topic of teacher education. It aims to reduce the gap between theory and practice in the professional training of primary teachers. The topis is quite original, being linked to the pedagogical research and methodology in a field that uses qualitative techniques such as semi structured interviews addressed to a small number of participants.
The text is clear and easy to read, but it requires a more detailed specification of the teachers' opinions about their training and the representations of their profession. The conclusions are quite consistent with the arguments presented, because they underline the need to link theory and practice through the shared experience.
I think that a deep comparison between this program and others initiatives may be more explanatory of the key challenges of teacher education.
The data results can be shown better, differenciating the answers under every theme.
Author Response

(The authors gave the same response as above.)

Reviewer 3 Report
I found some major methodological issues in this work.
There is a fundamental methodological issue about the type of research authors claim to be applying. The authors claim that their research is qualitative “because it aims to provide a description and interpretation of social phenomena” (lines 292-293). Social phenomena may be approached with a qualitative or a quantitative research. The type of research is not based on the nature of the data, but on the approximation and the data researchers decide to collect as well as the consequent analyses they choose to apply.
Please explain why you consider your data collection method to be a semi-structured interview and not an open-ended questionnaire. When the “second interview” as stated by the authors was missing, it seems to be a qualitative questionnaire.
Authors also need to explain further their data analyses procedure: did you use any qualitative research software or similar method?
Selection of participants is also not clear. Why and how did you select the stated sample?
Furthermore, if the aim of the paper is to show the kampus mengajar program to be valuable as opposed to other instruction other pre-service teachers receive, the need for a control-group is clear. If this is not the aim of the paper it should be stated more clearly.
In this sense authors mention in the discussion that “This research specifically highlighted the lack of relationship between the programs organized by the campus and the teacher’s field experience.” (lines 609-610). But the article shows only the kampus mengajar program, so that this kind of affirmations are difficult to maintain.
Other observations about you article are stated below.
Throughout the article authors state that teachers’ performance in Indonesia is very poor. However, they do not explain in what sense or how their lack of professionalism is shown.
Please review your keywords and include more relevant ones for online search (e.g. Kampus mengajar program is too specific).
Review language in:
- Page 2 lines 59-60: unfinished sentence.
-Page 4 line 162
- Page 5 line 245: repeated word “first
- Page 5 line 250: verb tenses
- Review the use of italics for “kampus mengajar program” as it is not consistent throughout your work.
- Try to avoid subjective an dcoloquial expressions such as “This is, therefore, a blessing for pre-service teachers” (line 405).
“real experienceincluding learning” (line 710)
Explain what your credit system implies (1 credit is how many hours of students' work?).
Explain sentence in page 2 line 70 "future leaders of the nation who are superior, moral, and ethical [19]." --> How are they superior?
Be sure to provide DOI or other link to cited articles, as some are very difficult to find (e.g. [36]).
Please specify all acronyms, as there are some that may not be known by all.
Explain “complicated bureaucracy” as used in Page 4 line 175.
Explain the term “partner schools” (page 4 line 190)
The “Author’s Role” paragraph does not belong in the corpus of the paper.
A table in the results section which summarizes the most relevant findings would help to gain an overview.
Explain how the mentors are selected as you state that “Therefore, those selected were based on expertise and not just a matter of seniority [11]” (line 650).
Author Response
Dear reviewers
Thank you for your response. We are very happy to receive feedback from reviewers described in our article. This shows that the reviewers are experts in this field. We are currently working hard to respond and revise our articles based on suggestions from reviewers. Please see the attachment for the revised manuscript.
King regards
Round 2
Reviewer 3 Report
It would have been helpful to receive a document that addresses the suggested changes and explains how they have been included in the new version of the manuscript. Please take this into account for further review processes.
Various improvements, especially concerning the research background, are observed.
Nevertheless, I still see an important issue concerning the research type. The type of research is not based on the nature of the data, but on the approximation and the data researchers decide to collect as well as the consequent analyses they choose to apply. To claim the use of a qualitative research because it helps to better understand human behavior is OK but not enough. That the research is qualitative or quantitative should be explained based on the kind of data collected and the analyses applied.
Furthermore, the first so called interview is an open-ended questionnaire as there is no contact between interviewer and interviewed. Only in the second round of data collection it may be called interview.
In your data analyses, explain how you addressed feasibility of your data.
Please review language throughout the paper, as there are still errors to be found.
I still do not understand the inclusion of the 2.5. paragraph about the authors' role.
Author Response
Thank you to the reviewers who whole heartedly help us improve the quality of this paper by volunteering their valuable time to read and provide suggestions for improvement. We are very happy with this response. We really appreciate all of your hard work and thank you for the suggestion. We believe that your expertise in reviewing our articles has contributed a lot in improving the quality of our articles. Although we admit that we have to work hard to be able to answer all responses, we are very grateful to receive your feedbacks, and we appreciate it highly. Thank you, we will try to address all the suggestions more specifically with a very satisfactory answer.
